# The Prognostic and Therapeutic Implications of the Chemoresistance Gene BIRC5 in Triple-Negative Breast Cancer

**DOI:** 10.3390/cancers14215180

**Published:** 2022-10-22

**Authors:** Getinet M. Adinew, Samia Messeha, Equar Taka, Karam F. A. Soliman

**Affiliations:** Division of Pharmaceutical Sciences, College of Pharmacy and Pharmaceutical Sciences, Institute of Public Health, Florida A & M University, Tallahassee, FL 32307, USA

**Keywords:** breast cancer, triple-negative breast cancer, surviving, BIRC5

## Abstract

**Simple Summary:**

TNBC is the most aggressive type of breast cancer affecting women worldwide, and chemoresistance poses a significant clinical challenge associated with a poor prognosis. The molecular mechanisms causing this treatment resistance in TNBC patients have not been extensively studied. This study was designed to find a prognostic biomarker that can accurately predict the patient’s disease status. We discovered that the chemoresistance gene BIRC5 could be a potential therapeutic target and a useful predictive biomarker for TNBC patients. We examined the expression of the genes that BIRC5 targets. Differentially expressed target genes were associated with carcinogenesis, tumor suppression, and cancer development. The most significant target genes were tumor oncogenes (TK1, KIF2C, UBE2C, AURKB) and tumor suppressors (CALCOCO1, CIRBP, KLHDC1, CBX7). It was concluded from this study that the findings might offer novel insights into TNBC chemoresistance and pinpoint key therapeutic targets, thereby assisting clinicians in developing alternative treatment options for TNBC patients.

**Abstract:**

Chemoresistance affects TNBC patient treatment responses. Therefore, identifying the chemoresistant gene provides a new approach to understanding chemoresistance in TNBC. BIRC5 was examined in the current study as a tool for predicting the prognosis of TNBC patients and assisting in developing alternative therapies using online database tools. According to the examined studies, BIRC5 was highly expressed in 45 to 90% of TNBC patients. BIRC5 is not only abundantly expressed but also contributes to resistance to chemotherapy, anti-HER2 therapy, and radiotherapy. Patients with increased expression of BIRC5 had a median survival of 31.2 months compared to 85.8 months in low-expression counterparts (HR, 1.73; CI, 1.4–2.13; *p* = 2.5 × 10^−7^). The overall survival, disease-free survival, relapse-free survival, distant metastasis-free survival, and the complete pathological response of TNBC patients with high expression of BIRC5 who received any chemotherapy (Taxane, Ixabepilone, FAC, CMF, FEC, Anthracycline) and anti-HER2 therapy (Trastuzumab, Lapatinib) did not differ significantly from those patients receiving any other treatment. Data obtained indicate that the BIRC5 promoter region was substantially methylated, and hypermethylation was associated with higher BIRC5 mRNA expression (*p* < 0.05). The findings of this study outline the role of BIRC5 in chemotherapy-induced resistance of TNBC, further indicating that BIRC5 may serve as a promising prognostic biomarker that contributes to chemoresistance and could be a possible therapeutic target. Meanwhile, several in vitro studies show that flavonoids were highly effective in inhibiting BIRC5 in genetically diverse TNBC cells. Therefore, flavonoids would be a promising strategy for preventing and treating TNBC patients with the BIRC5 molecule.

## 1. Introduction

Breast cancer (BC) is becoming the most common cancer worldwide [1]. BC has a poor prognosis, even though precision surgery and adjuvant systemic treatments such as chemotherapy, radiotherapy, and molecular targeted drugs significantly improve the overall outcome [2]. Because clinical, pathological, and genetic variables in personalized tumor therapy are restricted, an alternative procedure to predict prognosis and treatment response is urgently needed [3]. BC is a highly heterogeneous cancer concerning its clinical, histological, and molecular characteristics, treatment, and prognosis options. The expression of progesterone receptor (PR), estrogen receptor (ER), and human epidermal growth factor receptor 2 (HER2) is associated with the molecular subtype of BC [4]. The ER receptor, PR, and HER2 are not expressed in patients with TNBC. Therefore, TNBC is the most aggressive and invasive BC subtype. It accounts for about 15–20 percent of all incidences of BC [5,6]. TNBC has adverse prognostic aspects compared to other BC subtypes, such as a higher prevalence of visceral metastases, a shorter time between recurrences, and a higher nuclear grade [7]. The poor prognosis in TNBC patients stems from the limited treatment options available. Immunotherapy and chemotherapy (doxorubicin, paclitaxel, and cyclophosphamide) are the most often-used treatments for TNBC. However, this therapy has shown significant resistance and side effects [8,9]. Drug resistance is more frequently seen in TNBC patients than in people without TNBC [10,11]. Numerous genes and biological pathways have been associated with the various drug resistance mechanisms that have been found in studies of TNBC patients. Drug-mediated anti-tumor immune responses, for instance, are influenced by CD73 and CD133, and IMP3 controls the drug resistance proteins ABCG2 and heat shock factor 1 (HSF1), as well as the autophagy-related protein 7 (ATG7) [11,12,13,14]. Through the control of numerous biological processes in the human body, the PI3K/AKT/mTOR pathways have also been associated with drug resistance [15]. Recent studies have shown that co-opting of the LXRalpha: P-glycoprotein axis, a pathway highly targetable by therapies already utilized for the prevention and treatment of other diseases, is the cause of systemic chemotherapy failure in some TNBC patients [16]. These data suggest that specific genes are at responsibility for TNBC patients’ chemo- or other therapy resistance. However, there are no identifiable molecular targets in TNBC to underlie tailored treatment, and suppression of tumor cell apoptosis is one of the reasons for the failure of the used pharmacotherapy [17,18,19,20]. 

Many cancers, including BC, involve aberrant activation of proliferative pathways. Apoptosis inhibition is now well-recognized as a factor in the carcinogenic process [21]. Carcinogenesis is associated with an imbalance between cells’ apoptosis and proliferation stages. Many genes are either proapoptotic or antiapoptotic and regulate apoptotic signaling pathways. Regulation of cell death is essential for the preservation of homeostasis and healthy multicellular organism development. Inhibitors of apoptosis proteins (IAP) prevent apoptosis and necrosis, control the cell cycle and inflammation, and regulate cell death in various ways. IAP proteins are enticing candidates for the creation of innovative anticancer therapies because of their substantial capacity to be involved in cell death and enhanced expression in a range of cancer cell types. One of the essential genes in the IAP class that inhibits apoptosis is survivin (BIRC5) [22]. Baculoviral inhibitor of apoptosis repeat containing 5, BIRC5, a mitotic spindle checkpoint gene, has been demonstrated to play crucial roles in carcinogenesis by affecting cell division and proliferation and blocking apoptosis [23]. Studies from 420 patients with long-term clinical follow-up showed that BIRC5 was found in 378 (90%) of the 420 primary breast cancer cases, and levels were substantially related to negative hormone receptor status (*p* = 0.0028) [24]. A similar clinical study using tissue samples and patients found that out of 90 cases of TNBC, 19 (21.1%) tested negative for BIRC5 expression, and 71 (78.9%) tested positive [25]. Other studies reported that BIRC5 was over-expressed in 62 (45.6%) of 136 individuals with primary TNBC resected [26]. BIRC5 has a strong and independent association with TNBC patients and is a promising new target for future therapeutic strategies.

Treatment that targets BIRC5 has been recognized as a unique approach for numerous malignant tumors since BIRC5 is typically overexpressed in many malignancies [27,28]. In ovarian cancer, the cells’ ability to proliferate, migrate, and invade other tissues can be slowed down through both molecular suppressions by gene editing approaches and pharmacological inhibition by BIRC5 antagonists [28]. Recent research suggests that BIRC5 can regulate carcinogenesis. For instance, overexpression of BIRC5 has been associated with the development of BC and poorly differentiated tumors [29]. In in vitro and in vivo studies, penile cancer cell proliferation, migration, and invasion were all boosted by BIRC5 overexpression, while these same actions were reduced by BIRC5 knockdown. It also prolonged mice’s longevity and reduced PC xenograft tumors’ growth [30]. BIRC5 is significantly expressed in TNBC. Therefore, BIRC5 suppression reduces the growth of BC cells, suggesting that BIRC5 functions as a tumor driver [31]. Furthermore, research showed that the negative marker BIRC5 was associated with stage II/III BC that did not respond to neoadjuvant treatment [32]. 

Additionally, it has been discovered that BIRC5 expression confers resistance to chemotherapy and radiation. In experimental models, targeting BIRC5 increases the overall survival of patients [33,34]. An essential aspect of further study on the described genes and proteins is their potential use as a target of new strategies for targeted anticancer therapy because apoptosis inhibitor proteins and genes from the BIRC family control cell death and the signals of communication pathways. It is critical to precisely gauge the genes’ degree of expression in TNBC to comprehend the precise function of the BIRC family genes in TNBC [35]. 

In our laboratory, in vitro findings demonstrated that BIRC5 can be reduced in a major way in TNBC cells using natural products. Our studies employing the TNBC cell line MDA-MB-468 showed that rosmarinic acid significantly decreases BIRC5 mRNA expression [36]. Additionally, oleuropein, the most prominent polyphenol found in olive fruits and leaves, considerably reduced the mRNA expression level of BIRC5 in MDA-MB-231 TNBC cells [37]. Moreover, the natural polyphenol compound gossypol significantly decreased the mRNA expression level of BIRC5 in MDA-MB-231 and MDA-MB-468 TNBC cells [38]. In another recent study, the natural substance thymoquinone drastically lowered the mRNA expression level of BIRC5 in MDA-MB-231 TNBC cells [2]. Finding new compounds that target BIRC5 is crucial since BIRC5 is abundantly expressed in TNBC patients and, at the same time, resistant to chemotherapy and radiation therapy. 

We conducted the current study to ascertain the expression level of the BIRC5 gene in patients diagnosed with TNBC and, to some extent, compare the results with those of other BC subtypes to determine the role of the discussed genes as prognostic factors of TNBC. There have been a few studies that have focused on BIRC5 in various cancers, including BC. In the current study, we have revealed more comprehensive bioinformatics data to better understand the prognostic relevance of BIRC5 to TNBC. Chemoresistance, which generally affects patient treatment responses, has been associated with TNBC development and progression. Therefore, identifying a prognostic gene associated with chemoresistance, BIRC5, may lead to a new understanding of chemoresistance in TNBC and determining the prognosis of patients receiving chemotherapy, anti-HER2, and endotherapy. The chemoresistance gene BIRC5 was evaluated and validated in the current study as a tool for predicting the prognosis of TNBC patients and assisting in developing alternative therapies. To our knowledge, not enough research has been conducted to date to determine the expression level of BIRC5 in TNBC and chemoresistance profile for patients’ survival related to online database tools: UALCAN, The Breast Cancer Gene-Expression Miner v4.5 (bc-GenExMiner v4.8), TNM Plotter, TIMER2, and others. The current study aims to evaluate the expression of the antiapoptotic gene, BIRC5, in TNBC cells in relation to race, age, epigenetic alterations, hormonal status, and clinicopathological parameters. The information obtained will contribute to a better understanding of how to diagnose TNBC and treat TNBC and other cancers.

## 2. Materials and Methods

### 2.1. Gene and Protein Expression Analysis

#### 2.1.1. UALCAN Analysis 

The levels of BIRC5 mRNA transcripts in TNBC from the TCGA dataset were assessed using UALCAN (http://ualcan.path.uab.edu/analysis.html (accessed on 4 July 2022)), UALCAN is an open-access web portal that houses TCGA data such as gene expression, protein expression (CPTAC dataset), promoter methylation, mRNA expression, and clinicopathological information [39]. Users can utilize UALCAN to look up a gene of interest and compare its expression to clinicopathological characteristics. In this study, UALCAN was used to examine BIRC5 mRNA expression in relation to clinicopathological TNBC parameters (cancer stage, TNBC subtypes, TNBC race-based, metastasis, and grade).

#### 2.1.2. Bc-GeneExMiner v4.8 mRNA Expression Database

The Breast Cancer Gene-Expression Miner v4.8 (BCGEM) online data set was used to assess the expression and prognostic value of various apoptosis genes in TNBC. The online dataset is a statistical mining tool for published annotated breast cancer transcriptomic data, including DNA microarrays and RNA-seq. It can perform statistical analysis of gene expression, correlation, and prognosis (http://bcgenex.centregauducheau.fr (accessed on 4 July 2022)) [40,41,42]. The screening conditions set in this study are: “Analysis: expression-targeted,” “gene expression data: DNA microarrays;” Population: TNBC (IHC) and/or Basal-like (PAM50)”. 

#### 2.1.3. TNM Plotter 

The TNM Plotter (https://www.tnmplot.com (accessed on 2 August 2022)) is an online tool that allows for a real-time comparison of gene-expression changes in the tumor, normal, and metastatic tissues across many platforms. The program was utilized to examine BIRC5 expression using the TCGA datasets, with the Mann–Whitney test being employed to provide a direct comparison between tumor and normal tissues [43].

#### 2.1.4. TIMER2.0 

Gene connections and co-expression patterns of genes were assessed using TIMER2.0 across TCGA cancer categories. It was also used to compare the expression of *BIRC5* in TNBC with wildtype and mutant variants of major transcriptional factors linked to the start and progression of TNBC tumors. TIMER2.0 (http://timer.cistrome.org (accessed on 2 August 2022)) uses six state-of-the-art algorithms to offer a robust estimation of immune infiltration levels for The Cancer Genome Atlas (TCGA) or user-provided tumor profiles [44,45]. TIMER2.0 also includes modules for looking at the links between immune infiltrates and genetic or clinical characteristics, clinical outcomes, and cancer-related connections in the TCGA cohorts [44,45]. 

#### 2.1.5. Atlas of Human Proteins 

The Human Protein Atlas (HPA) was utilized to see if the *BIRC5* protein was expressed differently in normal breast and breast carcinomas (https://www.proteinatlas.org (accessed on 29 June 2022)) [46].

#### 2.1.6. The ROC Plotter

The ROC Plotter is an online tool for cancer research that validates predictive biomarkers at the transcriptome level (https://www.rocplot.org/ (accessed on 29 June 2022)) [47].

### 2.2. Analysis of Overall Survival

The predictive value of *BIRC5* expression in TNBC was determined using the online Kaplan–Meier plotter application (https://kmplot.com/analysis (accessed on 2 August 2022)). Kaplan–Meier survival plots were explicitly created to see if there was a link between *BIRC5* expression and TNBC patients’ overall survival (OS) [48,49]. Each percentile (of expression) between the lower and upper quartiles was computed to examine the prognostic significance of *BIRC5*, and the best-performing threshold was chosen as the final cutoff in a univariate Cox regression analysis. The Kaplan–Meier survival plot and the hazard ratio with 95% confidence intervals and log-rank *p*-value were calculated [48,49].

### 2.3. Methylation Analysis

MethSurv (https://biit.cs.ut.ee/methsurv (accessed on 29 June 2022)) was used to determine methylation status. Using the TGCA dataset, it is possible to do survival analysis for a CpG situated in or around the proximity of a query gene using this web-based application [50]. CpG visualization, pan-cancer methylation profile, differential methylation analysis, correlation analysis, and survival analysis are among the interactive and configurable features [51].

### 2.4. Gene Ontology and Pathway Analysis

LinkedOmics (http://www.linkedomics.org (accessed on 5 August 2022)) was used to visualize the gene ontology (GO) enrichment analysis results of the genes co-expressed with *BIRC5*. The LinkedOmics database combines multiomics and clinical data from the TCGA for 32 cancer types [27]. Within the TCGA KIRC cohort, the “LinkFinder” module was utilized to look for differentially expressed genes. Using the GO Biological Process and KEGG databases, the “LinkInterpreter” module was utilized to perform GSEA pathway enrichment analysis. Spearman’s correlation test was used to determine significance, using a *p*-value and false discovery rate (FDR). The projected *BIRC5* target genes were uploaded to the PANTHER (Protein Analysis Through Evolutionary Relationships) Classification System analysis online to find gene/protein networks that were over-represented within the gene collection. The PANTHER website (http://pantherdb.org/about.jsp) offers tools for functional analysis of gene or protein lists. Then, the lists were graphically evaluated using sortable functional classes and pie or bar charts or statistically using overrepresentation or enrichment tests [52,53,54,55].

### 2.5. Statistical Analysis

The TNM Plotter web analytic tool was used to undertake statistical analysis of tumor and normal-tissue gene expression. The Mann–Whitney U test was used to compare the normal and tumor samples. The Kaplan–Meier Plotter web analytic tool was also used to generate Kaplan–Meier survival plots with the number of people at risk, hazard ratio (HR), 95 percent confidence intervals (CI), and log-rank *p*–values. The *p*-value was set to < 0.05 to imply that there was a statistically significant difference in overall survival between the high-expression and low-expression groups

## 3. Results

### 3.1. BIRC5 mRNA and Protein Expression in Breast Carcinoma

In the breast cell cancer TCGA dataset, the expression of *BIRC5* transcripts is shown in Figure 1a. Compared to normal tissues, *BIRC5* mRNA expression was considerably increased (*p* = 6.03 × 10^−181^) with a mean fold change of 27.30. To verify the gene-expression data, we examined immunohistochemical pictures of human normal and BC tissues stained with antibodies produced against the *BIRC5* protein from the Human Protein Atlas (https://www.proteinatlas.org/ (accessed on 29 June 2022)). According to the captions of each image (found at https://www.proteinatlas.org/ENSG00000089685-BIRC5/pathology/breast+cancer#Intensity, breast carcinoma images demonstrated increased *BIRC5* protein expression. All normal tissues were reported as having “not detected” staining with “negative” intensities (Figure 1b,d), in contrast to BC tissues, which had “low” staining with “moderate” intensities (Figure 1c,e). This suggests that the *BIRC5* protein is upregulated in breast carcinomas. When *BIRC5* expression was examined in the heat map profile, TNBC (Figure 1f), HER2 (Figure 1g), and luminal (Figure 1h), as well as stage II of BC (Figure 1i), showed the highest levels of *BIRC5* expression, respectively.

### 3.2. BIRC5 Gene Expression Based on Hormone Status and BRCA1/2 Status

We examined the expression of *BIRC5* based on hormone status and *BRCA1/2* because *BIRC5* is heavily regulated by hormone and gene mutation. Compared to the ER^+^, the *BIRC5* expression was substantially upregulated in ER^−^ patients (Figure 2a, (*p* < 0.0001), and similarly, the expression was high in PR^−^ and HER^−^ compared to PR^+^ (Figure 2b) and HER2^+^ (Figure 2c), respectively. This finding was further confirmed by increasing the expression in basal type and TNBC patients compared to non-basal (Figure 2d) and non-TNBC patients (Figure 2e). Concerning the mutation of *BRCA1/2*, the mRNA *BIRC5* expression was higher in the mutant compared to the wild type (Figure 2f).

### 3.3. BIRC5 mRNA Expression in Association with the Clinicopathological Features of Triple-Negative Breast Cancer

We investigated *BIRC5* expression in several clinicopathological characteristics of breast cell cancer to characterize the protein. The mRNA expression of *BIRC5* was significantly higher in TNBC than in other subtypes (Figure 3a) and, at the same time, substantially higher in AA compared to Caucasian and Asian TNBC patients (Figure 3b). Stages two and three of the tumors had increased transcript expression of *BIRC5* significantly compared to normal, as well as stages one and four of the tumors, and *BIRC5* remained overexpressed (*p* < 0.001) throughout tumor stages (Figure 3c). Similar trends in nodal metastasis status were also observed, with *BIRC5* expression considerably upregulated in N0, N1, and N2 (Figure 3d) (*p* < 0.01). Additionally, there was a noticeable upregulation of *BIRC5* transcripts N3 metastasis, with *BIRC5* expression significantly increasing with increased nodal involvement (*p* < 0.01) and consequently higher metastatic status. The more aggressive subtype TNBC had the highest level of mRNA expression of *BIRC5*, which was overexpressed in BC subtypes (*p* < 0.001).

### 3.4. Overall Survival of Patients with Breast Cell Carcinomas as a Function of BIRC5 Expression Related to Different Treatments

Kaplan–Meier survival analysis was conducted to determine if increased expression of *BIRC5* is associated with modifications in overall patient survival. In individuals with breast cell carcinomas, higher expression of *BIRC5* mRNA was linked to a worse prognosis. A receiver operating characteristic (ROC) analysis was performed to verify KM survival observations and examine whether *BIRC5* expression might distinguish between high-expression and low-expression groups regarding survival and complete pathological response. Compared to the low-expression group, the high-expression group had a median survival of 68.04 months as opposed to 121.2 months in its counterpart (HR: 1.43; CI: 1.18–1.73; *p* = 2 × 10^−4^) (Figure 4a). In systematically treated patients, relapse-free survival was higher in low than in high BIRC5 expression. In comparison to the low-expression group, the high-expression group had an upper quartile survival of 37.25 months as opposed to 66.23 months (HR: 1.41; CI: 1.27–1.56; *p* < 2.9 × 10^−11^) (Figure 4b). Compared to the *BIRC5* low-expression systematically untreated patients, the high-expression group had a median survival of 37 months as opposed to 93.23 months of relapse-free survival (HR: 1.41; CI: 1.14–1.75; *p* = 0.0013) (Figure 4c). We look at the complete pathological response and relapse-free survival profile for all BC subtypes treated with chemotherapy agents (Taxane, Ixabepilone, FAC (Fluorouracil, Adriamycin, and Cytoxan), CMF (cyclophosphamide, methotrexate, and 5 fluorouracil (also known as 5FU)), FEC (5 fluorouracil (also known as 5FU), epirubicin, and cyclophosphamide), Anthracycline), and endotherapy drugs (Tamoxifen, Aromatase inhibitor), and any anti−HER2 therapy (Trastuzumab, Lapatinib). Patients receiving any form of chemotherapy (Figure 4d) or any form of endotherapy (Figure 4e) demonstrated a significant decline in the expression of *BIRC5*, which was supported by an increase in relapse-free survival. On the other hand, patients receiving anti−HER2 therapy did not display a significant difference (Figure 4f). In contrast, there was no discernible difference in the complete pathological response between patients receiving any anti–HER2 therapy (Figure 4g), endotherapy (Figure 4h), and chemotherapy (Figure 4i) and those who were not receiving any treatment. 

### 3.5. Complete Pathological Response of Patients with TNBC as a Function of BIRC5 Expression Related to Various Treatments

A Kaplan–Meier survival study was performed to see if an elevated expression of *BIRC5* is related to changes in the complete pathological response of TNBC patients. A worse outcome was associated with increased *BIRC5* mRNA expression in those with TNBC. A receiver operating characteristic (ROC) analysis was also performed to corroborate KM survival observations and examine whether *BIRC5* expression might distinguish between high-expression and low-expression groups regarding complete pathological response. Patients with TNBC who receive any chemotherapy (median of responder 369 (*n* = 1324) vs. non-responder 372 (*n* = 732) (Figure 5a) and anti-HER2 therapy 156 (*n* = 443) vs. non-responder 203 (*n* = 993) (Figure 5b) do not significantly vary from those who do not receive any form of treatment compared to all subtype of BC. Similarly, TNBC patients with nodal statuses of negative (median of responder 473 (*n* = 1034) vs. non-responder 375 (*n* = 1427) (Figure 5c) or positive (median of responder 530 (*n* = 1151) vs. non-responder 456 (*n* = 1732) (Figure 5d), as well as grade II (median of responder 193 (*n* = 1140) vs. non-responder 344 (*n* = 1732) (Figure 5e) and III (median of responder 432 (*n* = 1155) vs. non-responder 433 (*n* = 1427) (Figure 5f), have not demonstrated any difference in pathological response while receiving any form of chemotherapy or not (https://rocplot.org/site/treatment (accessed on 29 June 2022)).

### 3.6. Overall Relapse-Free Survival of Patients with TNBC as a Function of BIRC5 Expression Related to Chemotherapy

The relapse-free survival of TNBC patients receiving chemotherapy is also examined in more detail. The treated TNBC patients have not demonstrated a significant difference in the expression of *BIRC5*, as demonstrated by a non-significant difference in relapse-free survival compared to the non-treatment group (Figure 6a). The same holds for TNBC patients with grade II (Figure 6b) and III (Figure 6c) tumors, nodal positive (Figure 6d) and negative (Figure 6e), and estrogen-negative (Figure 6f) tumors. The luminal A subtype of BC has decreasing *BIRC5* expression on treated with any chemotherapy. Compared with non-responders, responders have decreased *BIRC5* expression with a median value of 111 vs. 320 (Figure 4d). 

### 3.7. Expression of BIRC5 in Relation to DMFS, OS, and DFS in TNBC Patients’ Prognostic Analysis

We further investigate the expression of *BIRC5* in relation to DFS (disease-free survival), OS (Overall survival), and DMFS (distant metastasis-free survival) in the three TNBC subtyped patients: C1: molecular apocrine tumors (or luminal androgen receptor); C2: Basal-like tumors with high levels of neurogenesis activity and immune suppressive cell infiltration; and C3: Basal-like tumors with an ineffective immune response, which is characterized by high levels of lymphocytes and plasma cells infiltrating the tumor, tertiary lymphoid structures, and upregulation of immune checkpoints. The DFS was more elevated in low *BIRC5* expression than in higher expression in the population C1 subtype of TNBC (*p* = 0.0065, HR = 2.61, CI = 1.31–5.21) (Figure 7a); the same is true for OS (*p* = 0.0146, HR = 3.44, CI = 1.28–9.27) (Figure 7b) and DMFS (*p* = 0.0148, HR = 3.71, CI = 1.29–10.65) (Figure 7c). Similar results were observed in TNBC population subtype C2 (Figure 7d–f). However, in population C, there was no discernible difference between low and high *BIRC5* expression (Figure 7g–i).

### 3.8. CpG Methylation’s Impact on BIRC5 Gene Expression in TNBC

We examined the methylation status of BIRC5 and compared it to mRNA expression since methylation plays a significant role in the regulation of genes. Compared to the low expression, high BIRC5 expression was associated with increased methylation, accompanied by decreased patient survival time. Compared to the control, there was a significantly decreased overall survival rate (*p* = 0.032) when promoter sites cg23302638 were hypermethylated (Figure 8a). The beta value indicates a DNA methylation level of 0 (unmethylated) to 1(fully methylated). Different beta value cut-off has been considered to indicate hypermethylation [Beta value: 0.7–0.5] or hypomethylation [Beta-value: 0.3–0.25]. In this study, the BIRC5 promoter region was substantially methylated, and hypermethylation was associated with higher mRNA expression with a Beta median value of 0.582 vs. 0.691 for control and tumor, respectively (*p* < 1 × 10^−12^) (Figure 8b). We analyze BIRC5′s methylation dependent on BC subtypes. The results demonstrated that, when compared to normal breast tissue, the TNBC subtype of BC had the highest Beta median value (0.631) (*p* = 0.012) (Figure 8c). Compared to AA and Asians, Caucasians have a greater beta value. Asians also have a greater beta value than AA. Age-related methylation has had a noticeable impact on the expression of BIRC5. Ages 81–100 have the highest Beta median value (0.697) compared to normal breast tissue, followed by ages 61–80 (0.691), 41–60 (0.689), and 21.40 (0.688). This result shows that BIRC5 methylation was directly correlated with age (Figure 8d). Finally, hypermethylation of BIRC5 was significantly associated with race, age, and hormone status. In terms of the patient’s race, Asians had the highest beta median value (0.696) compared to normal breast tissue, followed by Caucasians (0.691) and AAs (0.685) (Figure 8e).

### 3.9. Effect of Mutations in TP53, CDH1, RELN, PIK3CA, and MAP3K1 on BIRC5 mRNA Expression

We examined the expression of BIRC5 using wildtype and mutated forms of the following proteins: Tumor protein P53 (*TP53*), phosphoinositide-3-kinase, catalytic, alpha polypeptide (*PIK3CA*), Cadherin-1 (*CDH1*), Mitogen-activated protein kinase-1 (*MAP3K1*), Reelin (*RELN*), and Cytoplasmic dynein 2 heavy chain 1 (*DYNC2H1*). Dynein axonemal heavy chain 7 (*DNAH7*), FAT atypical cadherin 3 (*FAT3*), Bromodomain, and WD Repeat Domain Containing 1 (*BRWD1*), and Spectrin alpha, erythrocytic 1 (*SPTA1*) genes (Figure 9a–e). 

Figure 10 examines the relationships between various genes and *BIRC5* in different BC types. The degree of their association is indicated by Heatmap using the partial Spearman’s rho value that has been purity-adjusted. Spearman’s rho is a non-parametric test that assesses the degree of correlation between two variables; a positive correlation is indicated by a value of r = +ve, and a negative correlation is indicated by a value of r = −ve (TIMER2.0 (cistrome.org). 

All the mutated genes indicated above, except for *PIK3CA, CDH1*, and *MAP3K1,* showed significant overexpression of *BIRC5* in BC. While *MAP3K1* and *RELN* show a negative correlation with *BIRC5* in BRCA, *CDH1* positively relates with *BIRC5* in the BRCA–Luminal A subtype of BC. BRCA–Luminal A and B have shown a negative association with *MAP3K1*. The most mutated genes with their mean expression of mutant and wild type are demonstrated in Table 1. 

### 3.10. BIRC5 Pathway Enrichment, Target Gene Expression, and Target Gene Ontology in TNBC

Due to the scarcity of data on *BIRC5* target genes, we investigated whether the genes significantly associated with *BIRC5* expression were elements of over-represented pathways. We employed the gene set enrichment analysis (GSEA) method to find enriched pathways to examine the GO Biological Process and KEGG databases (Figure 11a–k). Figure 11a shows a volcano plot of the connected genes for *BIRC5* expression that are negatively (green line) and favorably (red line). Using the GO Biological Process database, Figure 11b shows the GSEA analysis of the pathways enriched among genes that co-express BIRC5 positively (blue bars) and negatively (orange bars). Only the pathways linked to the genes that BIRC5 inversely co-expressed were significantly enriched (*p* < 0.05, FDR < 0.05). The top seven pathways enriched among the inversely co-expressed genes are shown in enrichment plots in Figure 11c–j. The chromosome segregation (GO:0007059), organelle fission (GO:0048285), mitotic cell cycle phase transition (GO:0044772), spindle organization (GO:0007051), cytokinesis (GO:0000910), positive regulation of cell cycle (GO:0045787), and meiotic cell cycle (GO:0051321), were among these enriched pathways. The pathways related to the vascular endothelial growth factor receptor signaling pathway (GO:0048010) and negative regulation of cellular component movement (GO:0051271) are the most highly enriched among the positively correlated genes (Figure 11k). Similarly, we specifically used the gene set enrichment analysis (GSEA) method to explore KEGG databases to identify enriched pathways. Again, GSEA analysis of the KEGG database showed the pathways involved in the cell cycle (hsa04110), pyrimidine metabolism (hsa00240), and oocyte meiosis (hsa04114), and DNA replication (hsa03030) were primarily enriched among positively correlated genes (Figure 11l). Figure 11m–p shows a few of their respective enrichment plots. The AMPK signaling pathway (hsa04152) and most significant pathways related to Th17 cell differentiation (hsa04659) were the most significantly enriched pathways among inversely associated genes for genes positively co-expressed with BIRC5. A couple of their enrichment plots are shown in Figure 11h–r. 

We used the GSEA Molecular Signature Database to find 50 potential targets for *BIRC5* because it is a member of one of the most well-known transcription factor families. Figure 12a and b show a heatmap from the positive and negative correlation of the *BRC5* target gene in normal tissues and BC, respectively, using Ualcan.path.uab.edu/analysis. Compared to normal tissues, the most notable targets that had previously been linked to tumorigenesis, progression, or suppression of BC. *TIK1* (fold-change 1.22) (Figure 12c), *KIF2C* (fold-change 1.47) (Figure 12d), *UBE2C* (fold-change 5.25) (Figure 12e), and *AURKB* (fold-change 2.11) (Figure 12f) are among the positively correlated genes, while *CALCOCO1*(Figure 12g), *CIRBP* (Figure 11h), *KLHDC1* (Figure 12i), and *CBX7* (Figure 12j) are the among the negatively correlated genes. Figure 12k–o shows the correlation between *BIRC5* and selected target genes. Nevertheless, biological regulation, followed by cellular metabolic activities, constituted the biggest set of genes involved in the biological process (Figure 12p).

The best positive and negative correlation between *BIRC5* with a list of genes in TNBC by immunohistochemistry is demonstrated in Figure 13. The buttons with the color gradient display the colors of the correlation score codes (r), indicating the best correlation of *BIRC5* with a maximum of 50 genes. The top four best positive correlation genes with *BIRC5* are *MT2P1*, *RPL39P5*, *KIF4B*, and *CDCA5*, with correlation scores of 0.7578, 0.7512, 0.7007, and 0.6881, respectively. The bottom two best positive correlation genes with *BIRC5* are *DEPDC1* and *RAD54L*, with correlation scores of 0.5904 and 0.5882, respectively (Figure 13a). With correlation scores of −0.6726, −0.6119, −0.6093, and −0.6083, the top four genes with the best negative correlation to *BIRC5* are *CNN2P12*, *HNRNPA1P61*, *TMEM161BP1*, and *COX6B1P3*. *INPP4B* and LHFPL6 have correlation scores of −0.4335 and −0.4333, respectively, making them the bottom two best negative *BIRC5* correlation genes (Figure 13b).

We further examine the survival of TNBC patients among the positive correlation genes for *BIRC5*. Figure 14a demonstrates that high expression of *TK1* decreased the overall survival of TNBC patients compared with other subtypes and low/medium *TK1* expression for TNBC patients. In contrast, low *TTC28* expression from the inversely correlated genes is shown to decrease TNBC patients’ overall survival in Figure 14b. 

### 3.11. Pan-Cancer Analysis of the Expression of the BIRC5 Target Gene

We examined the expression of *BIRC5* in various cancers. Except for prostate adenocarcinoma, sarcoma, cutaneous melanoma, thyroid carcinoma, and thymoma, *BIRC5* was significantly elevated (*p* < 0.05) in all nineteen cancers based on the pan-cancer analysis results (Table 2).

## 4. Discussion

TNBC has historically had fewer therapeutic choices compared to other kinds of BC. Despite developing novel biologic and targeted agents, chemotherapy remains the backbone of treatment for TNBC even though chemoresistance and undesirable side effects are critical challenges. Finding predictive biomarkers that detect TNBC is, therefore, essential. The most frequently altered gene in TNBC is BIRC5, a family of BIRC (baculoviral inhibitors of apoptosis repeat-containing) proteins that could be used as a biomarker. These proteins also present a great potential to find molecular signatures that could function as prognostic or predictive markers for neoplastic diseases, including BC, given that malignancies are characterized by aberrant gene transcription. *BIRC5*, a mitotic spindle checkpoint gene, has been demonstrated to play crucial roles in carcinogenesis by affecting cell division and proliferation and blocking apoptosis [23]. In addition to being an essential protein molecule for controlling mitosis and apoptosis, *BIRC5* is also involved in pathogenic events [56]. *BIRC5* was shown to be overexpressed in tumor tissues. Since *BIRC5* is primarily expressed in tumor tissue, it may increase angiogenesis, promote cell division, and inhibit apoptosis [57]. High expression of *BIRC5* was associated with poor clinical outcomes in many cancers, including BC [32], hepatocarcinoma [58], pancreatic cancer [59], esophageal carcinoma [27], and neuroblastoma [60]. Additionally, circulating IgG antibodies generated from *BIRC5* may be used as a biomarker for the early detection of cervical cancer and malignant gliomas [61,62]. Moreover, the expression of *BIRC5* in bodily fluids may serve as a highly effective marker for the early detection and diagnosis of breast cancers [63]. Previous studies examined peripheral blood from BC patients to look for the presence of *BIRC5* in circulating breast tumor cells [54] and revealed that, out of 67 patients, 34 patients (50.7%) had breast tumor cells that expressed *BIRC5* in their peripheral blood samples but not in healthy samples [64]. Studies from 420 patients with long-term clinical follow-up showed that *BIRC5* was found in 378 (90%) of the 420 primary breast cancer cases, and levels were substantially related to negative hormone receptor status (*p* = 0.0028) [24]. A similar clinical study using tissue samples and patients found that out of 90 cases of TNBC, 19 (21.1%) tested negative for *BIRC5* expression, and 71 (78.9%) tested positive [25]. Other studies reported that *BIRC5* was over-expressed in 62 (45.6%) of 136 individuals who had their primary TNBC resected [26]. Studies reported that circulating BC cells expressing *BIRC5* might be related to several clinical characteristics, including tumor size, nodal involvement, HER2 expression, ER/PR status, and clinical stages of the illness [64,65]. It is now possible to suggest that *BIRC5* could serve as a potential target for the diagnostic and prognostic biomarker for the detection, diagnosis, or prognosis of breast tumor patients. Based on these results and the bioinformatics analyses outlined above, targeting *BIRC5* may be an essential strategy for treating TNBC.

In this study, we found that high levels of *BIRC5* mRNA expression predicted poor TNBC outcomes and had shorter OS, DFS, DMFS, and strong resistance to several treatment regimens. According to several studies, high *BIRC5* expression is positively correlated with a worse prognosis for survival and relapse from various cancer types [66,67]. In a cohort of individuals with acute myeloid leukemia, low *BIRC5* expression was also associated with statistically significant longer overall survival. The study also reveals a correlation between high *BIRC5* expression and prognosis [68,69]. Many cancers commonly express more *BIRC5*, which has been correlated to chemoresistance and a bad prognosis for cancer patients [70,71,72,73], which is consistent with our findings. In the current study, for all BC subtypes treated with any chemotherapy (Taxane, Ixabepilone, FAC, CMF, FEC, Anthracycline), any endotherapy (Tamoxifen, Aromatase inhibitor), and any anti-HER2 therapy (Trastuzumab, Lapatinib), have shown an increase in the complete pathological response, distant metastasis-free survival, overall survival, disease-free survival, and relapse-free survival of BC patients. In contrast, in TNBC patients, *BIRC5* was substantially associated with poor prognosis and resistance to these treatment regimens. The results demonstrate that TNBC patients receiving any form of chemotherapy or anti-HER2 therapy did not significantly differ from those receiving no treatment in terms of overall survival or complete pathological response. Treatment targeting *BIRC5* has been recognized as a unique approach for numerous malignant tumors because *BIRC5* is typically overexpressed in most cancers [27,28]. For instance, patients with head and neck cancer may respond more favorably to therapy if the nuclear export signal for *BIRC5* is deactivated [8]. In ovarian cancer, the cells’ ability to proliferate, migrate, and invade other tissues can be slowed down through both molecular suppressions by gene editing techniques and drug inhibition by *BIRC5* antagonists [28]. *BIRC5* was shown to be significantly expressed in the triple-negative subtype of BC, and *BIRC5* suppression reduced the growth of BC cells, suggesting that *BIRC5* functions as a tumor driver [31]. Furthermore, research showed that the negative marker *BIRC5* was associated with stage II/III BC that did not respond to neoadjuvant treatment [32]. Additionally, it has been discovered that *BIRC5* expression confers resistance to chemotherapy and radiation. In experimental models, *BIRC5* targeting increases survival [33]. 

The data collected in this investigation showed that *BIRC5* functioned consistently as a potential tumor-enhanced gene or as a prognostic marker. The *BIRC5* gene and protein expression are higher in BC, with TNBC being more prevalent than other BC subtypes (Figure 3a). In higher stages (II and III), mRNA levels are much higher, indicating that expression is grade- and stage-specific (Figure 3d). Studies show that *BIRC5* does not respond to neoadjuvant treatment in stage II/III breast cancer despite increased expression in stage II/II BC [32]. Given that overexpression is positively correlated to cancer development and aggressiveness, these data strongly imply that *BIRC5* may mediate carcinogenesis. Transcript expression did rise over time as metastatic nodal involvement grew (Figure 3c), and it was considerably higher in N0, N1, and N2. Additionally, lower patient overall survival was associated with increased transcription expression (Figure 4), indicating that *BIRC5* silencing may have a preventative effect on carcinogenesis in breast tissues. Similar previous studies on colorectal cancer found a substantial positive correlation between *BIRC5* expression and colorectal cancer stages, with *BIRC5* expression being higher in stages II/III [74]. Studies on Taiwanese BC patients showed that *BIRC5* was found in circulating tumor cells in their blood, and the results demonstrated that the expression of BIRC5 was significantly correlated with the size of the tumor, the histologic grade, the presence of lymph node metastases, and the TNM stage [75].

We examined whether epigenetic processes might be responsible for gene expression as *BIRC5* levels are noticeably elevated in TNBC. It was discovered that the *BIRC5* promoter and CpG foci were considerably hypermethylated. Additionally, there was a positive correlation between mRNA expression and hypermethylation, indicating that *BIRC5* expression may be partly caused by abnormal methylation. In contrast to the previous statement, it is evident that the amount of methylation raises the transcript expression of the gene [76]. However, based on the DNA methylation status, *BIRC5* expression might be negatively correlated with DNA methylation in TNBC. Furthermore, higher promoter methylation of *BIRC5* was observed in TNBC compared to normal tissues. We analyzed the effect of the most frequently changed genes on *BIRC5* expression because TNBC is also characterized by mutations in genes that are involved in chromatin remodeling and epigenetic controls, such as *TP53* and *BRCA1/2* [77]. 

*TP53* and *RELN* mutants both significantly increased *BIRC5* transcript levels, with *TP53* mutants having the most noticeable effects. By regulating the cellular cycle, chromatin remodeling, programmed cell death, and the immune response, *TP53* performs as a critical tumor suppressor. Indeed, TNBC and other aggressive tumors are associated with *TP53* gene loss function [78]. The *RELN* gene was identified as a crucial tumor suppressor gene and was found to be epigenetically silenced. *RELN* gene loss function in several tumors, including pancreatic [79], gastric [80], and BC [81], A high risk of recurrence of hepatocellular carcinoma is related to decreased expression of *RELN*, which was also associated with increased migratory ability, shorter survival, and a poor prognosis [82]. On the other hand, *BIRC5* expression is reduced in *CDH1, MAP3K1*, and *PIK3CA* mutants. It is believed that histone modification, like DNA methylation, is crucial to the transcriptional control of *BIRC5* because all these mutant genes are involved in epigenetic controls and chromatin modeling.

*BIRC5* is a putative transcription factor; thus, we used the GSEA Molecular Signature Database to find possible target genes. In BC, we examined the expression of the genes that *BIRC5* targets. Differentially expressed target genes were associated with carcinogenesis, tumor suppression, and cancer development. The most significant target genes were putative tumor oncogenes/promoters (*TK1, KIF2C, UBE2C, AURKB*) and potential tumor suppressors (CALCOCO1, CIRBP, KLHDC1, CBX7). Thymidine kinase 1 (TK1) is an enzyme in the DNA repair pathway that restores thymidine for use in DNA synthesis and DNA damage. *TK1* is crucial for cell repair after DNA damage in addition to DNA synthesis. Recent research suggests that over-expressed *TK1* promotes cancer cell invasion, proliferation, and progression [83]. In addition to being a byproduct of cancer cell processes, overexpression of *TK1* might also result from selection mechanisms that promote the growth of cancer cells. In lung adenocarcinoma and BC cell lines, TK1 has been found to support tumor growth; bioinformatic evidence points to a similar role for *TK1* in adrenocortical carcinoma and prostate cancer patients [84,85,86]. A recent study found a correlation between an elevated risk of BC and a mutation in the calcium-binding and coiled domain 1 (*CALCOCO1^R12H^)* in human BCs [87]. The cold-shock protein known as Cold-inducible RNA-binding protein (*CIRBP)* contains an RNA-binding motif activated by various stresses. Target mRNA is regulated post-transcriptionally by *CIRBP*, which is necessary for controlling DNA repair and cell proliferation. Additionally, it has been documented that *CIRBP* plays a critical role in several human disorders, including cancer and inflammatory disease. Despite being primarily thought of as an oncogene, *CIRBP* may potentially play a part in tumor suppression [88]. A significant prognostic marker and therapeutic target for cancer treatment may be *CIRBP*.

Finally, we examined *BIRC5* expression in other cancers and determined whether *BIRC5* might be a widespread tumor marker (Table 2). In this study, except for prostate adenocarcinoma, sarcoma, cutaneous melanoma, thyroid carcinoma, and thymoma, *BIRC5* expression was elevated (*p* < 0.05) in all cancer types, demonstrating that *BIRC5* expression is altered across a variety of tumor cohorts and may thus play a general role in the development and spread of cancer.

Previous in vitro findings in our lab on *BIRC5* demonstrated that it is a promising target for treating TNBC patients using natural products. The results obtained in the current study complement our earlier research. In our laboratory, in vitro findings demonstrated that BIRC5 can be reduced in a major way in TNBC cells using natural products. Our studies employing the TNBC cell line MDA-MB-468 showed that rosmarinic acid significantly decreases BIRC5 mRNA expression [36]. Additionally, oleuropein, the most prominent polyphenol found in olive fruits and leaves, considerably reduced the mRNA expression level of BIRC5 in MDA-MB-231 TNBC cells [37]. Moreover, the natural polyphenol compound gossypol significantly decreased the mRNA expression level of BIRC5 in MDA-MB-231 and MDA-MB-468 TNBC cells [38]. In another recent study, thymoquinone drastically lowered the mRNA expression level of BIRC5 in MDA-MB-231 TNBC cells [2]. These date suggest that *BIRC5* may not only operate as an oncogene but also as a promising predictive biomarker and possible therapeutic target in cancer [34]. Treatment targeting *BIRC5* has come to be recognized as a promising therapeutic approach due to the high activation of *BIRC5* during carcinogenesis in various cancer types [33]. The precise expression pattern, possible role, prognostic significance, and race-based expression of *BIRC5* in BC, however, are all yet completely unknown. However, these data suggest that *BIRC5*-targeting drugs may be recommended for treating TNBC or preventing chemoresistance.

## 5. Summary

In summary, *BIRC5* belongs to the *BIRC* subfamily, which is expressed differently in various malignancies; nonetheless, a poor prognosis is mainly associated with high expression in BC, including TNBC. *BIRC5* is essential for the development of tumors when considered as a whole, and changes in expression may function as a biomarker. *BIRC5* enhances cell proliferation, mitosis, and motility and inhibits apoptosis through the inhibition of caspases 3/9 [89,90,91]. It has a dismal prognosis for cancer patients and is significantly expressed in many tumors. Here, we used several online data tools to investigate the *BIRC5* gene expression patterns in TNBC and BC subtypes. We demonstrate that greater *BIRC5* RNA levels are associated with DNA amplification. Our findings further confirm that *BIRC5* expression levels are higher in cancer than in normal tissue, which is consistent with earlier findings [3]. We observe a correlation between increased *BIRC5* expression and poor OS in TNBC, and the results show that *BIRC5* expression in TNBC is relevant for prognosis. Figure 15 summarizes the association between patient prognosis, treatment outcomes, clinicopathological features, and the expression level of *BIRC5* in TNBC.

## 6. Strength, Limitation, and Implication of the Study

The study provided comprehensive clinicopathological information on TNBC individuals with high expression of *BIRC5* and might help researchers work into *BIRC5* as a potential target for the treatment and prevention of TNBC. Additionally, compared to previous studies, this research gives more emphasis on the chemoresistance-related effect of BIRC5 on a patient’s life. It employs a variety of online bioinformatics data tools to investigate the predictive relationship between *BIRC5* and TNBC, which is crucial to confirm the consistency of the findings and the importance of this gene as a potential target molecule. 

Our study’s primary flaw is that it relied on information from public databases that was not independently confirmed in clinical trials used in our investigation. Prospective clinical trials must validate these findings. Additionally, as a potential biomarker for early diagnosis, clinical trials should be used to discover the precise concentration of BIRC5 to diagnose TNBC as well as its sensitivity and specificity. The clinical characteristics of the human tissues employed in this study are not completely explained. Additionally, more research is needed on the methods through which BIRC5 contributes to the development of TNBC, as well as on the roles that BIRC5 plays in this process through in vivo and in vitro experiments. Although the potential predictive biomarker of TNBC, BIRC5, was examined in the current study, there are not enough samples from the database to prevent bias. Our findings may offer essential data and pathways about TNBC for a deeper understanding of the molecular process of TNBC carcinogenesis. 

Precision medicine and monitoring accurate biomarkers that can identify cancers in their early stages, including TNBC, will likely be the foundation of cancer patient management in the future. This study will open up therapists to concentrate on target-based treatments. Finding efficient biomarkers and treatment strategies will be crucial for developing precision medicine. Future research into potential biomarkers may help develop strategies to avoid developing drug resistance and improve the effectiveness of targeted therapy for cancer types. Therefore, this multi-omics-driven study may play a role in converting the existing treatments, thereby finding potential agents targeting *BIRC5,* which enhance patient quality of life and bring about a cure for TNBC and other cancers.

## 7. Conclusions

The most frequent cancer diagnosed in women worldwide is breast cancer. The most aggressive type of BC is TNBC, which does not express HER2, progesterone, or estrogen receptors. Several dysregulated genes are responsible for the development of TNBC. The dysregulated gene with the highest level of expression in TNBC is BIRC5. According to studies, on average, BIRC5 was expressed highly in 45 to 90% of TNBC patients. According to several in vitro studies and data from public databases, BIRC5 is not only abundantly expressed but also contributes to resistance to chemotherapy, anti-HER2 therapy, and radiotherapy. Not just TNBC patients but practically all other cancer types exhibit considerable expression of BIRC5. BIRC5 could be a potential biomarker for diagnosing TNBC and an optional target for efficient treatment. Focusing on the nutraceutical would be an alternate strategy for preventing and treating TNBC patients. Some in vitro studies employing natural products in genetically diverse TNBC cells show promising effects on inhibiting BIRC5. Patients with TNBC and other malignancies could be prevented and treated with the development of therapeutic agents targeting BIRC5.

BIRC5 gene expression is associated with more advanced stages of cancer, tumor grades, levels of metastasis, and aggressive subtypes of BC, including TNBC. As a result, *BIRC5* represents a risk factor for tumor growth that may be predicted, and it may also be a potential therapeutic target in the treatment of TNBC. We anticipate that *BIRC5* mRNA expression profiling in biopsies will be most useful as a diagnostic tool for screening and identifying people at a higher likelihood of developing the aggressive BC subtypes-TNBC, even though further large-scale validation is necessary. Comprehensive analysis is essential to obtain and validate *BIRC5* gene targets and transcription regulators, notably in TNBC, given the absence of knowledge surrounding the transcriptional targets of *BIRC5*. Our research showed that the *BIRC5* gene was expressed more highly in BC patients compared to healthy people. These online databases also revealed that, when compared to the corresponding normal tissues, *BIRC5* was markedly elevated in TNBC. Additionally, we examined how dysregulation of *BIRC5* in BC is caused. After analyzing the DNA methylation status in TCGA, we observed that *BIRC5* expression was directly correlated with DNA methylation and that BC tissues had higher levels of *BIRC5* promoter methylation. These findings suggested that DNA methylation may play a significant role in the deregulation of *BIRC5* in BC. According to functional enrichment analysis, genes co-expressed with *BIRC5* were strongly related to biological regulation, metabolic process, and cellular response to stimuli. Additionally, methylation and *BIRC5* expression levels were substantially correlated. This finding establishes the basis for the subsequent investigations into the modulation of TNBC by *BIRC5*, which could develop more potent targeted cancer treatments.

## Figures and Tables

**Figure 1 cancers-14-05180-f001:**
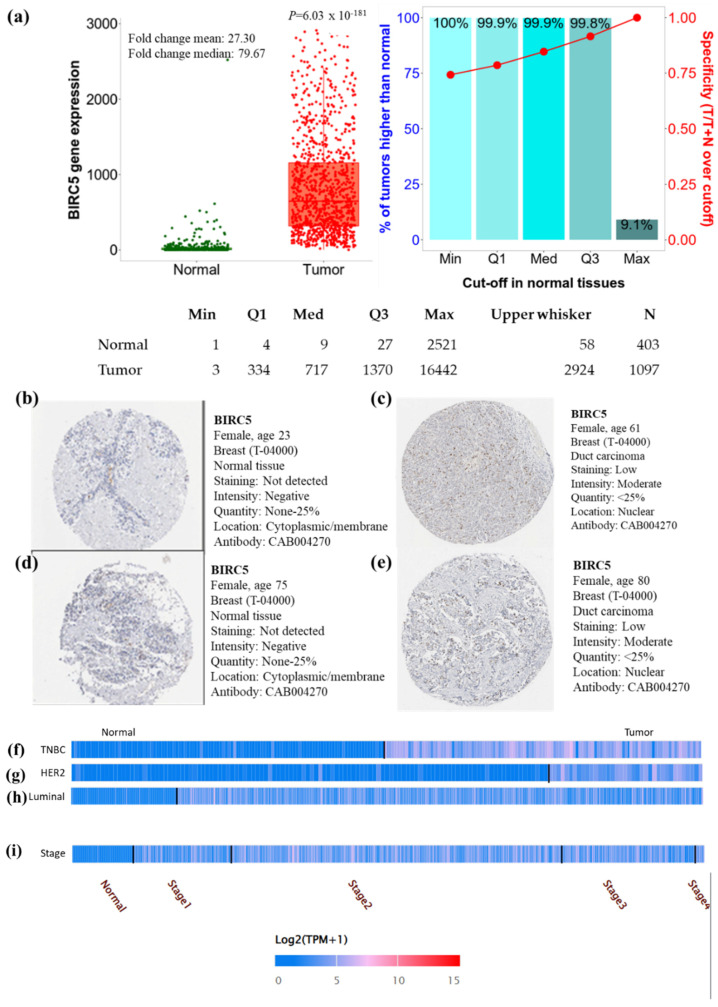
Breast cell cancer *BIRC5* transcript and protein expression. (**a**) Plots were created using TNMplot analysis of RNAseq data using TCGA datasets (https://tnmplot.com (accessed on 2 August 2022)). In comparison to normal tissues (*n* = 403), *BIRC5* transcripts were considerably upregulated (fold change = 27.3) in malignancies (*n* = 1097). (**b**,**d**) Immunohistochemistry showing the expression of the *BIRC5* protein normal breast tissue with anti-*BIRC5* antibody (CAB004270) and (**c**,**e**) BC with anti-*BIRC5* antibody (CAB004270), respectively, as shown on the Human Protein Atlas website (https://www.proteinatlas.org (accessed on 29 June 2022)). We found representative tumor images at https://www.proteinatlas.org/ENSG00000089685-BIRC5/pathology/breast+cancer#Intensity (accessed on 29 June 2022) and for normal breast tissue at https://www.proteinatlas.org/ENSG00000089685-BIRC5/tissue/breast (accessed on 29 June 2022). (**f**–**i**) demonstrates the heat map expression profile of TNBC, HER2+, and Luminal and pathological stage of BC.

**Figure 2 cancers-14-05180-f002:**
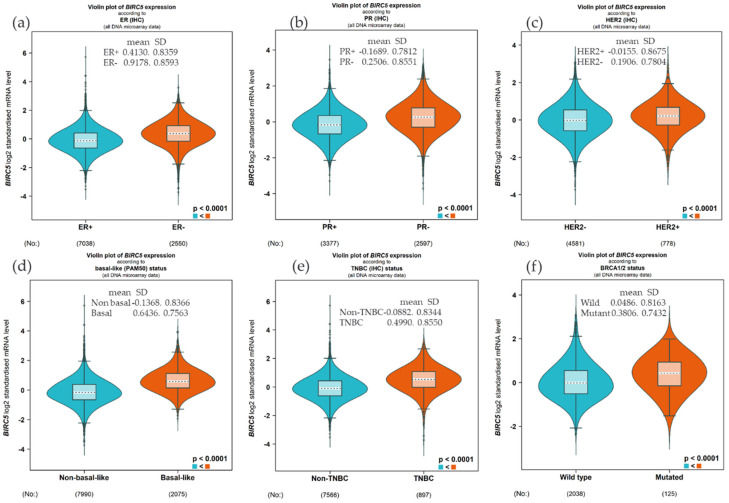
Expression of BIRC5 based on hormone status and BRCA1/2 mutation. Compared to the ER+, the BIRC5 expression was substantially upregulated in ER^−^ patients (**a**), and the expression was high in PR^−^ and HER^−^ compared to PR^+^ (**b**) and HER2^+^ (**c**). Compared to non-basal (**d**) and non-TNBC patients (**e**), the expression was high in basal and TNBC patients. The mRNA BIRC5 expression was higher in the BRCA1/2 mutant compared to the wild type (**f**).

**Figure 3 cancers-14-05180-f003:**
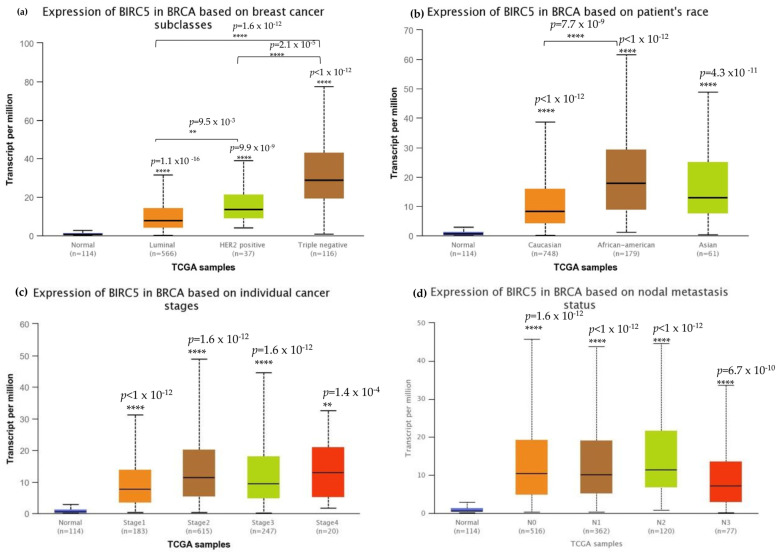
Breast cell carcinoma transcript expressions across cancer stages. (**a**) Expression of BIRC5 transcripts in different breast carcinoma histological grades. All BC subtypes of BIRC5 transcripts increased, with TNBC showing the highest expression. (**b**) BIRC5 expression among different patients’ races, in which AA show high expression. (**c**) In comparison to normal tissues, BIRC5 transcripts were markedly upregulated at all tumor stages. (**d**) Metastatic tissues express BIRC5 transcripts. Greater nodal involvement was associated with higher BIRC5 transcript levels (*p* < 0.01). (**d**) RNAseq data from TCGA datasets were analyzed using UALCAN to produce all graphs. ** *p* < 0.01, and **** *p* < 0.0001.

**Figure 4 cancers-14-05180-f004:**
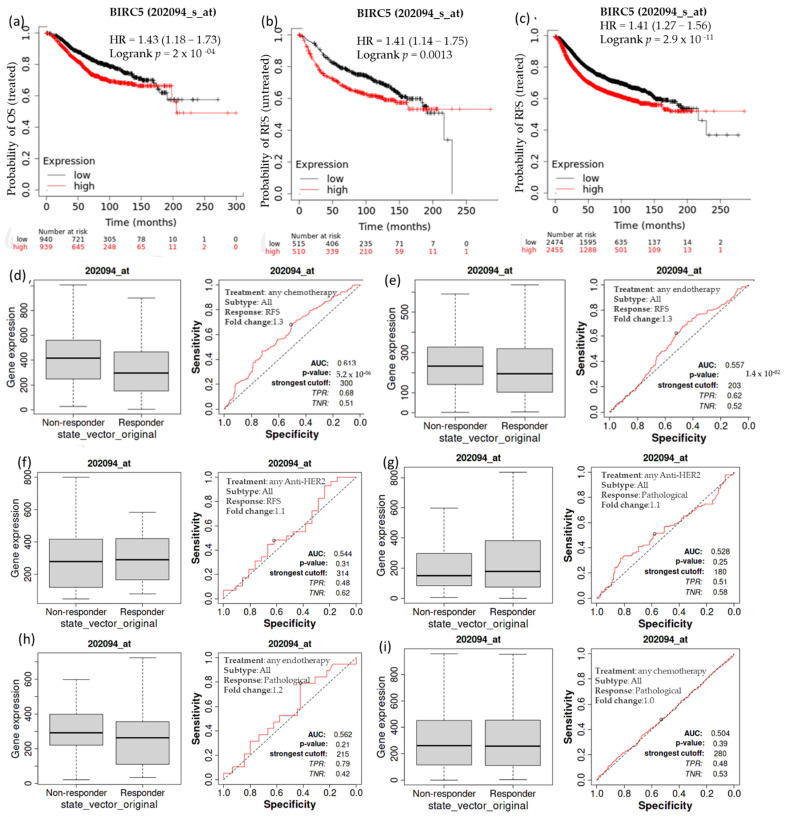
The relationship between BIRC5 expression, relapse-free survival, and complete pathological response in BC patients. Compared to the low-expression group, the high-expression group had a median survival of 68.04 months as opposed to 121.2 months HR, 1.43; CI: 1.18 –1.73; *p* = 2 × 10^−4^) (**a**). Compared to the low-expression group, the high-expression group had an upper quartile survival of 37.25 months as opposed to 66.23 months (HR:1.41; CI: 1.27−1.56; *p* < 2.9 × 10^−11^) (**b**). In comparison to the BIRC5 low-expression systematically untreated patients, the high-expression group had a median survival of 37 months as opposed to 93.23 months of relapse-free survival (HR:1.41; CI: 1.14−1.75; *p* = 0.0013) (**c**) Patients receiving any form of chemotherapy (**d**) and any form of endotherapy (**e**) demonstrated a significant decline in the expression of BIRC5, but patients receiving anti-HRE2 therapy did not display a significant difference (**f**). No discernible difference in the complete pathological response between patients receiving an anti–HER2 treatment (**g**), endotherapy (**h**), and chemotherapy (**i**) and those who were not receiving any treatment. OS-overall survival, RFS-relapse-free survival.

**Figure 5 cancers-14-05180-f005:**
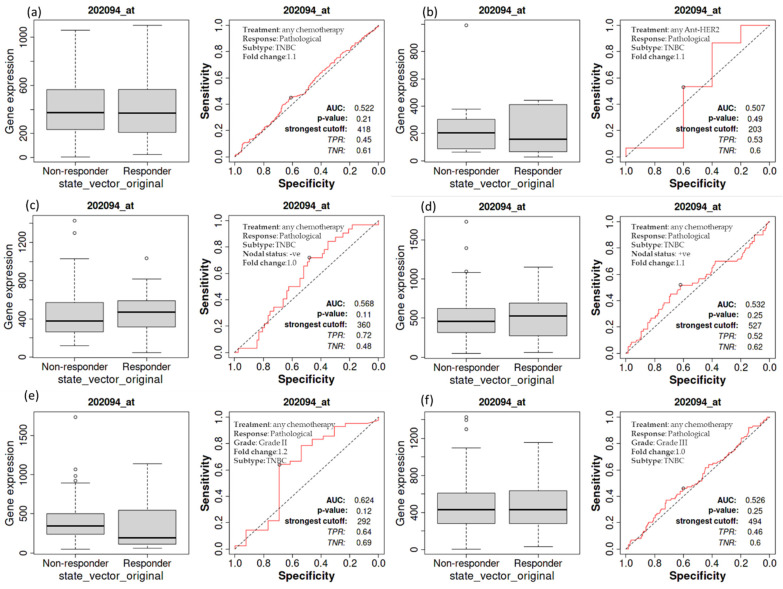
The relationship between BIRC5 expression and complete pathological response in TNBC patients. TNBC patients receiving any chemotherapy (**a**) and anti_HER2 therapy (**b**) have no significant difference from those patients not receiving any type of treatment. TNBC patients with nodal status negative (**c**) and positive (**d**), as well as with grade II (**e**) and III (**f**), treated with any type of chemotherapy.

**Figure 6 cancers-14-05180-f006:**
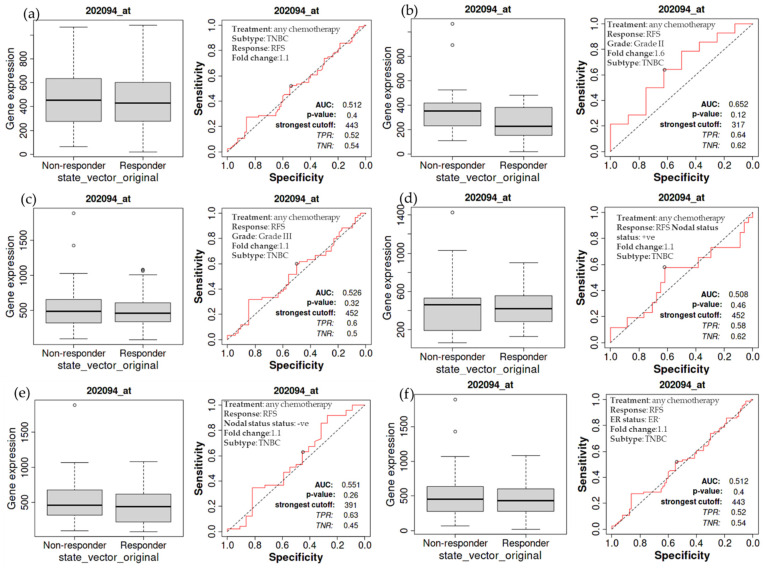
Relapse-free survival of patients with TNBC as a function of BIRC5 expression. Any chemotherapy-treated TNBC patients did not show a significant difference in BIRC5 expression, as shown by a non-significant difference in relapse-free survival (**a**). This also applies to TNBC patients with cancers of grade II (**b**), grade III (**c**), nodal positive (**d**), nodal negative (**e**), and estrogen negative (**f**) TNBC patients.

**Figure 7 cancers-14-05180-f007:**
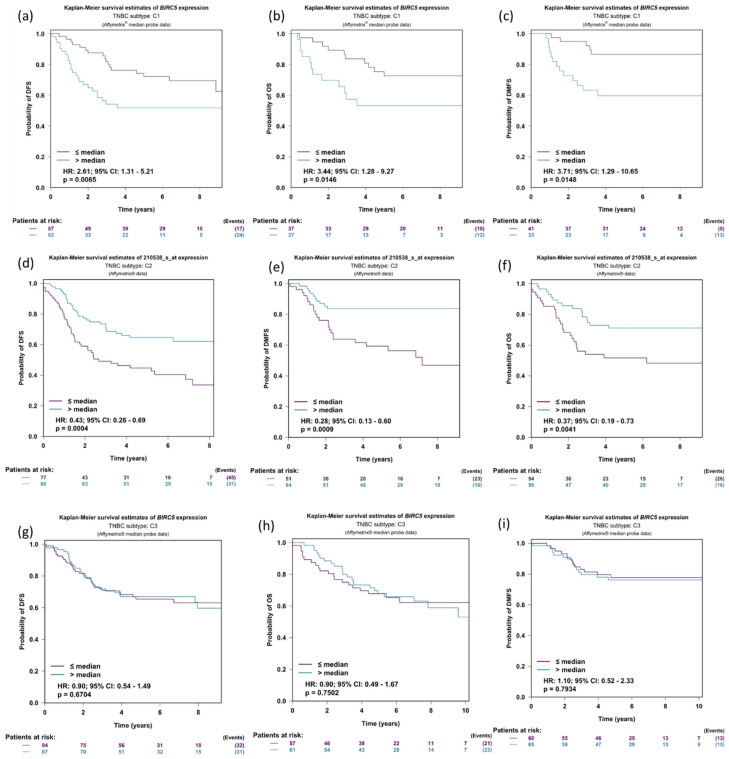
Expression of BIRC5 in relation to DMFS, OS, and DFS in TNBC patients. Expression of BIRC5 in population C1 (**a**–**c**), C2 (**d**–**f**), and C3 in respect to DFS (disease-free survival), OS (overall survival), and DMFS (distant metastasis-free survival) (**g**–**i**). For the BIRC5 gene, distributed in accordance with the splitting criterion chosen for all-event criteria (DMFS, OS and DFS), univariate Cox proportional hazards analysis and Kaplan–Meier curves are carried out (TNBC subtypes prognostic analysis module|bc-GenExMiner (unicancer.fr), accessed on 4 July 2022)).

**Figure 8 cancers-14-05180-f008:**
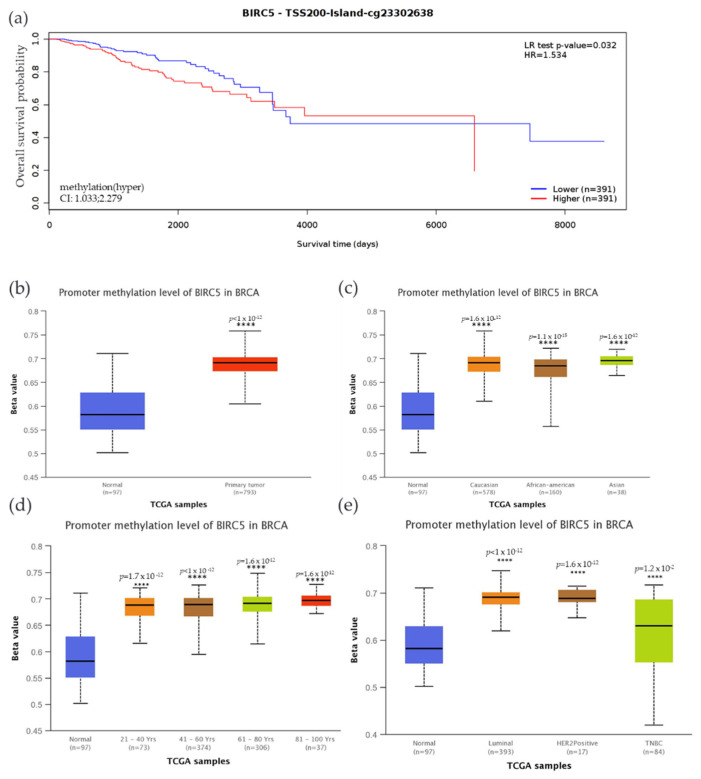
Methylation status of the BIRC5 gene. (**a**) BIRC5 methylation is associated with a decrease in the survival of BC patients. (**b**) The BIRC5 gene was also considerably hypermethylated (*p* < 1 × 10^−12^) in comparison to the normal tissues. Analysis of the relationship between the methylation status at CpG foci and the expression of the BIRC5 showed that hypermethylation was associated with hormone status (**c**), age (**d**), and race (**e**). **** *p* < 0.0001.

**Figure 9 cancers-14-05180-f009:**
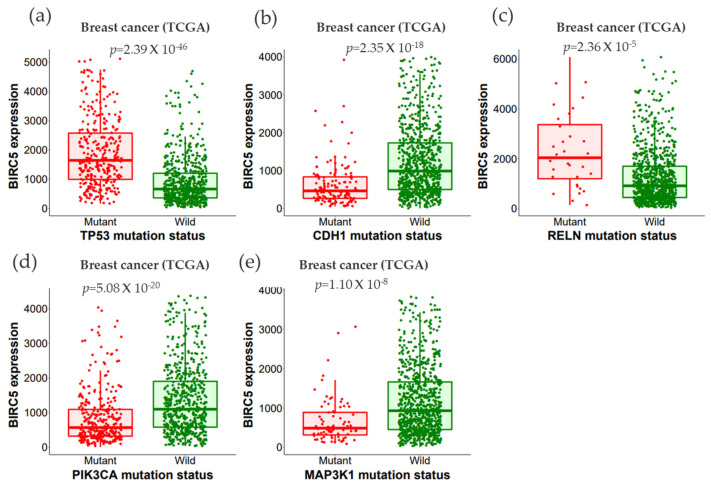
The relationship between *BIRC5* expression and essential gene mutations. (**a**–**e**) show the expression of *BIRC5* with the mutation status of the most frequently changed genes (*TP53, CDH1, RELN, PIK3CA,* and *MAP3K1*). Violin plots were generated using TIMER2.0 (http://timer.cistrome.org. *TP53*, tumor protein P53; *PIK3CA*, phosphoinositide-3-kinase, catalytic, alpha polypeptide; *CDH1*, cadherin-1; MAP3K1, mitogen-activated protein kinase 1; *RELN*, reelin; *DNAH7*, dynein axonemal heavy chain 7; *FAT3*, FAT atypical cadherin 3, *BRWD1*, Spectrin alpha, erythrocytic 1 (*SPTA1*), and *BRWD1*, Bromodomain and WD Repeat Domain Containing 1 gene.

**Figure 10 cancers-14-05180-f010:**
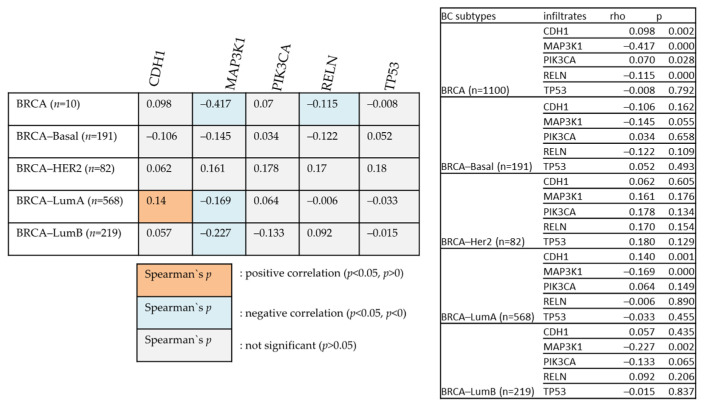
The positive and negative correlation between the mutated genes with BIRC5 in BC subtype patients.

**Figure 11 cancers-14-05180-f011:**
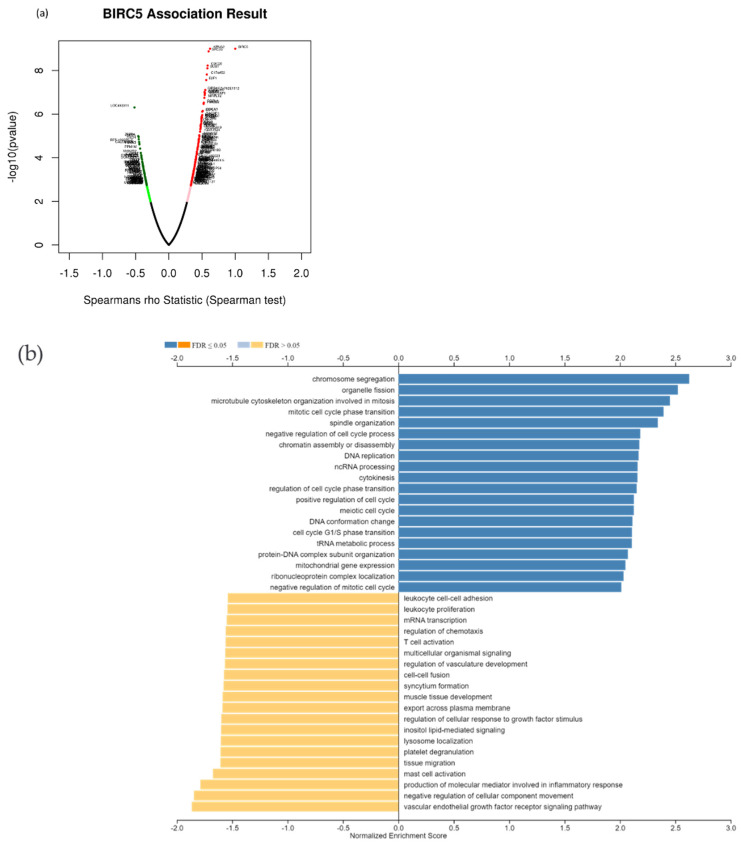
BIRC5 pathway enrichment, target gene expression, and target gene ontology. (**a**) Volcano plots of the genes with BIRC5 expression that are negatively and positively correlated. Genes that were negatively associated with BIRC5 expression are represented by the green section of the volcano curve, whereas positively correlated genes are shown by the red section of the curve (accessed on LinkedOmics:: Volcano Plot). GSEA GO (**b**) genes that co-express BIRC5 in a biological process. Blue bars show pathways that are highly enriched in positively linked genes, with dark blue bars denoting pathways with FDR < 0.05 and light blue bars representing pathways with FDR > 0.05. Orange bars show routes that are more likely to contain genes that are inversely correlated, with light orange bars showing pathways with FDR > 0.05. Inversely associated genes were overrepresented in the topmost significant pathways, as demonstrated by the enrichment plots in (**c**–**k**). (**g**) GSEA KEGG for BIRC5 and co-expressed genes KEGG database enrichment plots. (**l**–**r**) show the topmost significant pathways that are over-represented among negatively correlated genes, www.linkedomics.org/lo_batchfile/qindex_gsea.php?fn=120514 (accessed on 5 August 2022).

**Figure 12 cancers-14-05180-f012:**
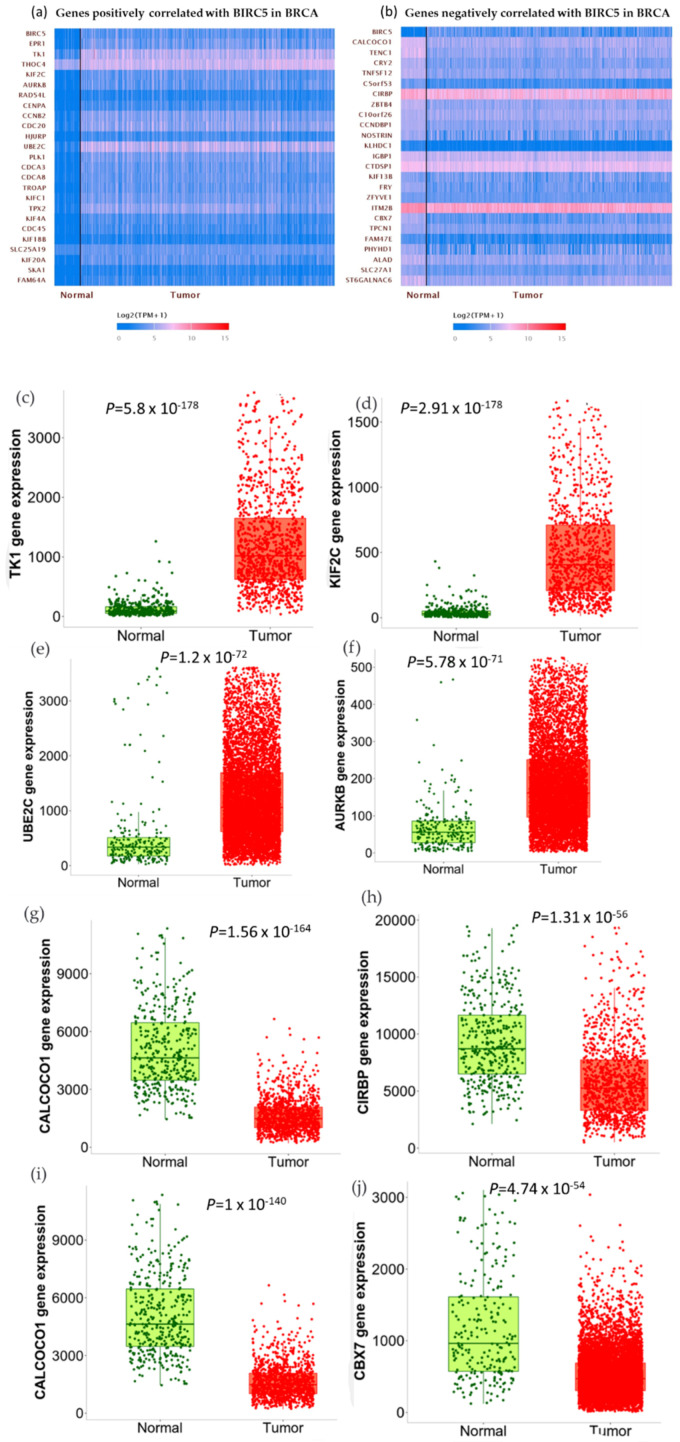
BIRC5 targets gene expression in breast cancer. (**a**,**b**) A heatmap created by UALCAN analysis shows how the anticipated BIRC5 target genes are expressed (Ualcan.path.uab.edu/analysis (accessed on 4 July 2022)). (**c**–**j**) The most notable targets implicated in cancer suppression, progression, or carcinogenesis, as well as those previously identified, had significant alterations in BC compared to normal. (**k**–**o**) The correlation between BIRC5 and target genes. (**p**) A pie chart featuring biological processes in which target genes are components, generated using PANTHER (http://www.pantherdb.org (accessed on 1 July 2022)).

**Figure 13 cancers-14-05180-f013:**
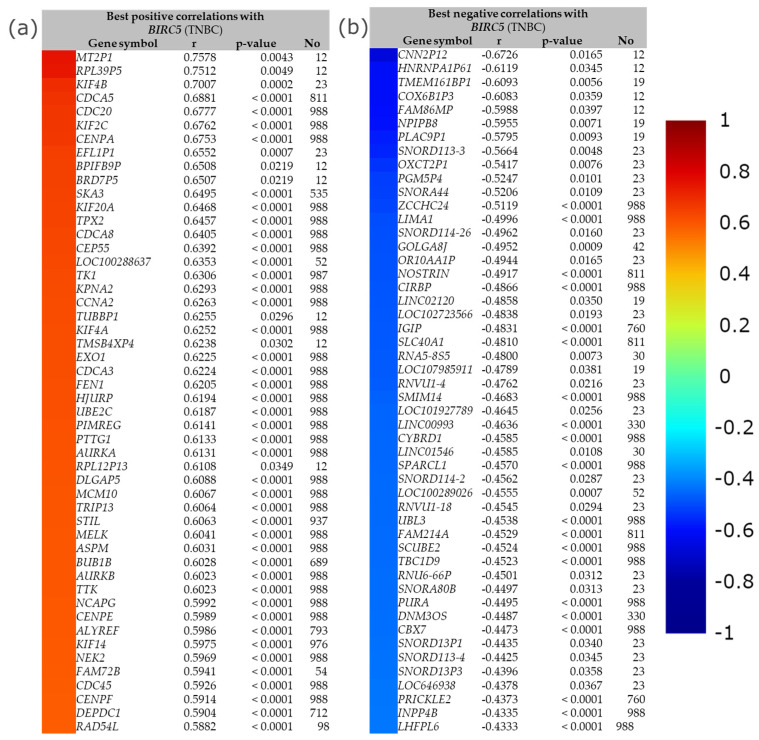
The best positive and negative correlation between BIRC5. (**a**) The best positive correlation with BIRC5 and (**b**) the best negative correlation with BIRC5. Gene expression correlation analyses (all DNA microarray data) are available in the gene correlation exhaustive analysis module|bc-GenExMiner (unicancer.fr), accessed on 5 July 2022.

**Figure 14 cancers-14-05180-f014:**
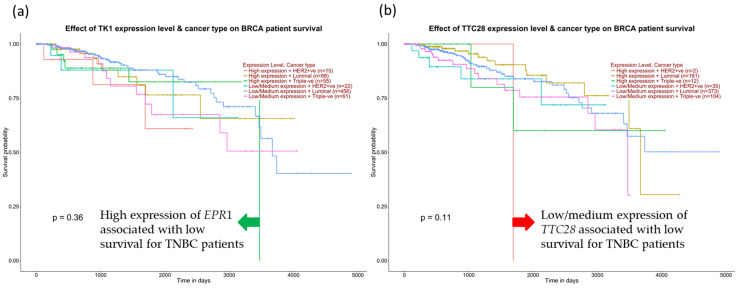
Expression of positive and negative correlation for *BIRC5* and TNBC patients’ survival. (**a**) High expression of *EPR1* from positively correlated genes decreases TNBC patients’ survival. (**b**) Low expression of *TTC28* from the inversely correlated genes drops TNBC survival, accessed from the UALCAN database (Ualcan.path.uab.edu/analysis) on 3 July 2022.

**Figure 15 cancers-14-05180-f015:**
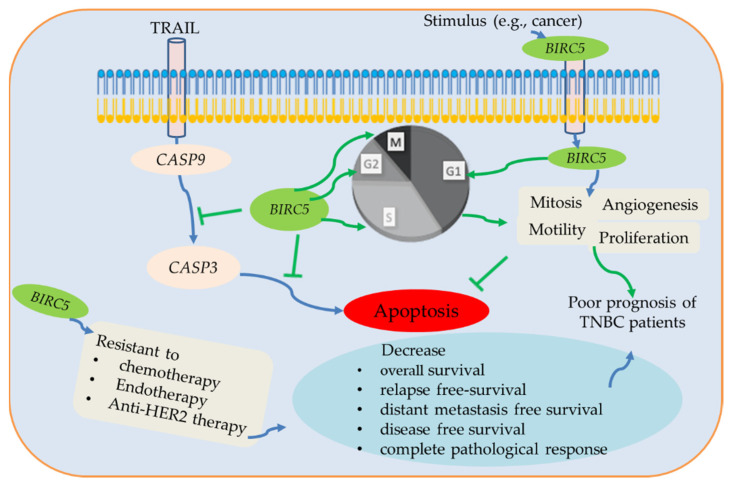
The potential BIRC5 pathways and their association with TNBC. BIRC5 inhibits apoptosis by preventing CASP9/3 activation and promoting cell division by regulating the various stages of the cell cycle. Additionally, BRIC5 is resistant to chemotherapy, endotherapy, and anti-HER2 therapy, which reduces complete pathological response and relapse-free survival of TNBC patients. Due to all of these conditions, BIRC5 expression has been associated with a poor prognosis for TNBC patients, suggesting that it may be a potential therapeutic target for the treatment and prevention of TNBC patients.

**Table 1 cancers-14-05180-t001:** The most frequently altered genes that significantly changed the expression of BIRC5. The data were accessed on muTarget|muTarget.

Mutation of	Mean Expression (Mutant)	Mean Expression (Wild)	Number of Mutants	Number of Wild	FC (Mutant/Wild)	Direction	*p*-Value
TP53	2041.5	1016.21	336	643	2.01	up	2.39 × 10^−46^
PIK3CA	887.52	1604.73	323	656	1.82	down	5.08 × 10^−20^
CDH1	659.35	1484.4	138	841	2.27	down	2.35 × 10^−18^
MAP3K1	708.36	1426.81	80	899	2	down	1.10 × 10^−8^
RELN	2328.66	1335.64	32	947	1.74	up	2.36 × 10^−5^
DYNC2H1	2477.62	1330.61	32	947	1.86	up	6.09 × 10^−5^
FAT3	2105.73	1333.39	44	935	1.58	up	1.97 × 10^−4^
BRWD1	2280.61	1346.15	23	956	1.69	up	4.10 × 10^−4^
DNAH7	2015.08	1350.45	26	953	1.49	up	9.11 × 10^−4^
SPTA1	1959.44	1333.58	54	925	1.47	up	1.17 × 10^−3^

**Table 2 cancers-14-05180-t002:** The expression pattern of BIRC5 in various cancers. The threshold for statistical significance was *p* < 0.05.

Tumor	Normal	Change	Significance (*p*-Value)
BLCA.Tumor (*n*=409)	BCLA.Normal (*n*=19)	Upregulation	5.35 × 10^−11^
BRCA.Tumor (*n* = 1097)	BRCA.Normal (*n* = 114)	Upregulation	<1 × 10^−12^
CESC.Tumor (*n* = 305)	CESC.Normal (*n* = 3)	Upregulation	1.62 × 10^−12^
CHOL.Tumor (*n* = 36)	CHOL.Normal (*n* = 9)	Upregulation	3.42 × 10^−8^
COAD.Tumor (*n* = 286)	COAD.Normal (*n* = 41)	Upregulation	1.62 × 10^−12^
ESCA.Tumor (*n* = 184)	ESCA.Normal (*n* = 11)	Upregulation	<1 × 10^−12^
GBM.Tumor (*n* = 156)	GBM.Normal (*n* = 5)	Upregulation	<1 × 10^−12^
HNSC.Tumor (*n* = 520)	HNSC.Normal (*n* = 44)	Upregulation	1.62 × 10^−12^
KICH.Tumor (*n* = 67)	KICH.Normal (*n* = 25)	Upregulation	4.13 × 10^−2^
KIRC.Tumor (*n* = 533)	KIRC.Normal (*n* = 72)	Upregulation	1.62 × 10^−12^
KIRP.Tumor (*n* = 290)	KIRP.Normal (*n* = 32)	Upregulation	1.11 × 10^−16^
LIHC.Tumor (*n* = 371)	LIHC.Normal (*n* = 50)	Upregulation	1.62 × 10^−12^
LUAD.Tumor (*n* = 515)	LUAD.Normal (*n* = 59)	Upregulation	1.62 x 10^−12^
LUSC.Tumor (*n* = 503)	LUSC.Normal (*n* = 52)	Upregulation	<1 × 10^−12^
PAAD.Tumor (*n* = 178)	PAAD.Normal (*n* = 4)	Upregulation	2.70 × 10^−1^
PRAD.Tumor (*n* = 497)	PRAD.Normal (*n* = 52)	Upregulation	7.22 × 10^−11^
PCPG.Tumor (*n* = 179)	PCPG.Normal (*n* = 3)	Upregulation	<1 × 10^−12^
READ.Tumor (*n* = 166)	READ.Normal (*n* = 10)	Upregulation	9.85 × 10^−7^
SARC, Tumor (*n* = 260)	SARC.Normal (*n* = 2)	Upregulation	1.22 × 10^−1^
SKCM.Tumor (*n* = 472)	SKCM.Normal (*n* = 1)	Upregulation	N/A
THCA.Tumor (*n* = 505)	THCA.Normal (*n* = 59)	Upregulation	8.80 × 10^−1^
THYM.Tumor (*n* = 120)	THYM.Normal (*n* = 2)	Upregulation	7.16 × 10^−1^
STAD.Tumor (*n* = 415)	STAD.Normal (*n* = 34)	Upregulation	2.54 × 10^−12^
UCEC.Tumor (*n* = 546)	UCEC.Normal (*n* = 35)	Upregulation	1.62 × 10^−12^

## Data Availability

All data generated or analyzed during this study are included in this published article.

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
