# Peer review of "The Prognostic and Therapeutic Implications of the Chemoresistance Gene BIRC5 in Triple-Negative Breast Cancer"

_cancers, 2022, doi:10.3390/cancers14215180_

Round 1

Reviewer 1 Report

In this manuscript, the authors described the prognostic and therapeutic role of BIRC5 in breast cancer, especially in triple-negative breast cancer (TNBC). The authors evaluated both the mRNA and protein expression of BIRC5 in breast cancer (BC), and found BIRC5 is over-expressed in BC, especially in TNBC. Next, they examined the expression of BIRC5 on different characteristics, including ER, PR, HER2, BRCA1/2 status, and tumor stage. They evaluated the relationship between OS and pCR of patients to different treatments and BIRC5 expression. They further investigated the expression of BIRC5 in relation to DMFS, OS, and DFS in TNBC patients. To explore what could affect the expression of BIRC5, they examined the CpG methylation, mutations of TP53, CDH1, RELN, PIK3CA, and MAP3K1. With GSEA analysis, they evaluated the pathways enriched among genes that co-express BIRC5 positively and negatively and identified several targets. Lastly, they performed pan-cancer analysis of the expression of the BIRC5 and found BIRC5 was elevated in all 19 cancers. Overall, the authors suggested the prognostic role of BIRC5 in TNBC and summarized the possible role of BIRC5 in TNBC. 

Author Response

No comments

Reviewer 2 Report

The authors have presented a well-studied work on gene BIRC5 and its role in TNBC as studied from patient samples. This work does correlate to their earlier reports in which they have observed similar phenomenon in TNBC cell lines such as MDA-MB-468, MDA-MB-231 etc. The work is indeed very interesting. There are couple of minor comments I would like the authors to address.

 1. Could the vertical axis of graphs a, b and c in Figure 4 be renamed as Survival or relapse-free survival. ‘Probability’ here seems to be misleading or not appropriate. Similarly, in Figure 7 and some other figures.

2. Similar to BIRC5 gene studied by the authors here, there are some recent very interesting work which have reported genes LXRalpha, ITGA7 and SEMA6D which have role in development of chemoresistance in triple negative breast cancer patients and also verified in cell lines (https://doi.org/10.1038/s41388-021-01720-w). The authors should include these in the introduction or discussion section.

Author Response

Dear Editor:

We are pleased to resubmit the revised version of the Manuscript ID:  cancer- 1972104 "The Prognostic and Therapeutic Implications of the Chemoresistance Gene BIRC5 in Triple-Negative Breast Cancer". We appreciate the reviewers' constructive criticisms, questions, and comments, and we have addressed each of their concerns as outlined below.

Reviewer 2

The authors have presented a well-studied work on gene BIRC5 and its role in TNBC as studied from patient samples. This work does correlate to their earlier reports in which they observed a similar phenomenon in TNBC cell lines such as MDA-MB-468, MDA-MB-231, etc. The work is indeed very interesting. I want the authors to address a couple of minor comments.

  1. Could the vertical axis of graphs a, b, and c in Figure 4 be renamed as Survival or relapse-free Survival. 'Probability' here seems to be misleading or not appropriate, similarly to Figure 7 and some other figures.
    • Response: corrected as suggested, included on
      1. page 2, lines 331-332, as the "OS and RFS" probability on the vertical axis for figure 4 a, b, and c.
      2. page 13, lines 404-405 as the probability of "DFS, probability of DMFS, and probability of OS" on the vertical axis for Figure 7.
  • page 14, lines 434-435 as "overall survival probability" on the vertical axis for Figure 8.
  1. Similar to the BIRC5 gene studied by the authors here, some recent very interesting work reported genes LXRalpha, ITGA7, and SEMA6D, which have a role in developing chemoresistance in triple-negative breast cancer patients and also verified in cell lines (https://doi.org/10.1038/s41388-021-01720-w). The authors should include these in the introduction or discussion section.
  • Response: included as suggested, included on page 2, lines 66-69 as" Recent studies have shown that co-opting of the LXRalpha: P-glycoprotein axis, a pathway highly targetable by therapies already utilized for the prevention and treatment of other diseases, is the cause of systemic chemotherapy failure in some TNBC patients."
